# Willingness to Pay for HPV Vaccine among Women Living with HIV in Nigeria

**DOI:** 10.3390/vaccines11050928

**Published:** 2023-05-03

**Authors:** Folahanmi T. Akinsolu, Olunike Abodunrin, Ifeoluwa E. Adewole, Mobolaji Olagunju, Aisha O. Gambari, Dolapo O. Raji, Ifeoma E. Idigbe, Diana W. Njuguna, Abideen Salako, Oliver C. Ezechi

**Affiliations:** 1Department of Public Health, Faculty of Basic Medical and Health Sciences, Lead City University, Ibadan 212102, Nigeria; 2Nigerian Institute of Medical Research, Lagos 100001, Nigeria; 3Lagos State Health Management Agency, Lagos 100001, Nigeria; 4Department of Epidemiology and Biostatistics, School of Public Health, Nanjing Medical University, Nanjing 210008, China; 5School of Nursing, Dedan Kimathi University of Technology, Nyeri 10100, Kenya

**Keywords:** HPV, cervical cancer, HPV vaccine, women, HIV

## Abstract

Background: Human papillomavirus (HPV) is responsible for most cervical cancer cases globally, with women living with HIV having a higher risk of persistent HPV infection and HPV-associated disease. The HPV vaccine is a promising tool to reduce cervical cancer rates, but its uptake among women living with HIV in Nigeria is unknown. Methods: A facility-based, cross-sectional survey was conducted with 1371 women living with HIV to assess their knowledge of HPV, cervical cancer, and the HPV vaccine as well as their willingness to pay for the vaccine at the HIV treatment clinic at the Nigerian Institute of Medical Research, Lagos. To identify factors associated with the willingness to pay for the HPV vaccine, multivariable logistic regression models were developed. Results: This study found that 79.1% of participants had not heard of the vaccine, and only 29.0% knew its efficacy in preventing cervical cancer. In addition, 68.3% of participants were unwilling to pay for the vaccine, and the average amount they were willing to pay was low. Knowledge of HPV, the HPV vaccine, and cervical cancer and income were factors associated with the willingness to pay for the vaccine. Health workers were the primary source of information. Conclusions: This study highlights the lack of knowledge and low willingness to pay for the HPV vaccine among women living with HIV in Nigeria and emphasizes the importance of improving education and awareness. Factors associated with the willingness to pay, such as income and knowledge, were identified. Practical strategies, such as community outreach and school-based education programs, could be developed to increase vaccine uptake. Further research is needed to explore additional factors influencing the willingness to pay.

## 1. Background

With more than 600,000 newly reported cases and about 300,000 fatalities from cervical cancer in 2020, it significantly threatens public health worldwide [1,2]. Low- and middle-income countries (LMICs) account for more than 80% of the mortality and incidence, particularly in sub-Saharan Africa, and have the highest death rates. [2,3]. The cervical cancer occurrence rate in Nigeria is 250 per 100,000 women of all ages, an endemic figure that indicates an insurmountable public health problem. It predominantly occurs in women living with HIV (WLWH) [4,5]. HIV-positive women have higher rates of both infection and mortality due to cervical cancer than their HIV-negative counterparts [4]. Cervical cancer and HIV are closely intertwined, and the number of patients with comorbidities continues to increase. Most individuals contract HPV at some point, typically shortly after beginning sexual activity [1,6]. HPV 16 and 18 are responsible for most quasi-cervical HPV-associated malignancies and 70.0% of cervical cancer. Although infection with HPV has been a significant contributory factor for cervical cancer, additional indicators of risk include early sexual contact, early marriage (before age 20), having more than one partner, sexual activity without protection, long-term hormonal contraception, a significant number of conceptions, tobacco use, and unsanitary habits [2,7].

According to the World Health Organization, vaccinating girls aged 9–14 against HPV can prevent at least one third of all HPV-related cancers in Africa [8,9]. Many high-income countries have now included HPV vaccination for adolescent girls in their routine immunization schedule [10,11]. In sub-Saharan Africa, the five countries with the highest rates of cervical cancer deaths emphasize the need for increased uptake of HPV vaccination in the region [8]. Hence, nations must execute efficient, cost-effective, and enduring HPV distribution tactics that align with their healthcare frameworks to attain optimal coverage. Since 2012, various international and domestic initiatives have implemented experimental programs to identify the most effective approaches for enhancing the distribution of HPV vaccines in LMICs [12,13].

Despite the success of cervical cancer screening programs in developed countries, screening for cervical cancer remains unpopular in Nigeria [14,15,16] as a result of the lack of an organized national screening program, and HPV screening is primarily opportunistic, with an estimated coverage rate of around 8.70% [15,17]. Nevertheless, two varieties of vaccines have proven effective in preventing cancer of the cervical cavity in Nigeria: the bivalent vaccine from GSK (Cervarix ™, Philadelphia, PA, USA) and Merck & Co. Inc.’s quadrivalent HPV vaccine (Gardasil, Boston, MA, USA). These vaccines are highly effective in preventing persistent HPV infection and subsequent precancerous lesions caused by two types of HPV (16 and 18), which are responsible for about 70.0% of cervical cancer cases worldwide [15,17,18,19].

Considering all-female populations is crucial in achieving the global goal of cervical cancer elimination. WLHIV are particularly important, as they have higher rates of persistent HPV infection and HPV-associated disease. Furthermore, countries such as Nigeria, with increased rates of HIV, also tend to have low rates of cervical cancer screening, making simple and affordable vaccine schedules critical [4]. Studies have shown that the HPV vaccine is safe, immunogenic, and effective in WLWH, with the best immune responses observed in those with undetectable HIV viral loads.

Introducing HPV vaccines in developing countries has been a significant challenge due to high prices. The Vaccine Alliance (Gavi) and its partners aim to provide the poorest countries with access to a sustainable supply of new and underused vaccines, including HPV vaccines, for as low as USD 4.50 per dose (NGN 2072.25) [18,20]. Gavi assists with HPV demonstration initiatives and the nationwide implementation of HPV vaccinations, contingent upon the nation’s established capacity to administer vaccinations to girls in their adolescence [15,18]. In Africa, Rwanda, Tanzania, South Africa, and Senegal are among the countries with the HPV vaccine in their national programs for immunization, following successful pilot projects [12,21]. A few other countries, including Nigeria, have ongoing pilot programs [1,20,22]. Gavi’s continuous provision of vaccine encouragement in Nigeria encompasses a range of vaccines, including pentavalent, pneumococcal conjugate, yellow fever, meningitis A, and measles vaccines. In addition, Gavi provides financial assistance to bolster the healthcare structure and enhance the vaccination program’s capacity [15]. Presently, individuals incur personal expenses when acquiring vaccines against HPV, as they are not included within the roster of vaccines that are provided free of charge through the National Immunization Program (NIP) in Nigeria. This exclusion is due to the potential exacerbation of the government’s already limited spending on health if free HPV vaccination were to be implemented. The implementation of HPV vaccination necessitates the creation of a novel vaccine distribution mechanism to effectively target adolescent females, given the absence of an established infrastructure to facilitate this undertaking, in addition to the associated expenses. Thus, despite the assistance provided by Gavi, substantial funding is required for the delivery of HPV vaccination to the community.

Research has indicated that impediments to the adoption of HPV immunization include the cost; parental inclination to finance the vaccination for their female offspring; insufficient awareness, particularly among caregivers of the intended demographic and particularly in lower-middle-income nations such as Nigeria; negative beliefs; and opposing attitudes [23,24]. In Nigeria, many studies that have examined the knowledge and parental acceptance of the HPV vaccine have reported low levels of expertise [25] and high levels of vaccine acceptability [7,15,25,26,27]. Therefore, this study aimed to ascertain the variables linked to the inclination to pay for the HPV vaccine among WLWH in Nigeria. This study also sought to assess the knowledge of HPV, cervical cancer, and the HPV vaccine among this population.

## 2. Materials and Methods

### Study Design and Setting

This research used a cross-sectional survey at the HIV treatment clinic at the Nigerian Institute of Medical Research (NIMR), Lagos State. The institute is a preeminent institution dedicated to researching diseases with significant public health implications in Nigeria. The institute presently offers a comprehensive range of HIV care, treatment, and support services to a population exceeding 20,000 individuals, with women constituting 62.9% of the total.

Study population: The study population included women of known HIV status aged 18 years and above that were receiving treatment and were eligible for cervical cancer screening at the NIMR clinic, Lagos State. Women who were unable to provide informed consent were excluded from the study.

Sampling method and sample size determination: The sampling method for this study was convenience sampling, a type of non-probability sampling technique. The sample size was calculated according to the formula N = Zα^2^P (1 − P)/d^2^, where Zα is the Z statistic for a 95% confidence level, N is the sample size, P is the prevalence of WLWH willing to pay for the HPV vaccine, and d is the precision. The population proportion of WLWH in Nigeria that were willing to pay for the HPV vaccine was 50%, with a 95% confidence level and a 5% margin of error; the minimum required sample size that was calculated was approximately 385 participants. However, with an increase in the precision estimate and accounting for potential non-responses or missing data, the sample size was increased to 1371 participants using a margin of error of approximately 2.5% with a 95% confidence level.

Data collection tool: The questionnaire was pretested for reliability and validity using a pilot study with a small sample of participants before its administration in the main study. The study used an interviewer-administered questionnaire to gather data on various demographic variables of the participants, including their age, level of education, primary occupation, and average monthly per capita income. In addition, the study questionnaire collected data on HPV, cervical cancer, and HPV vaccination knowledge. For the evaluation of participants’ level of comprehension regarding the ailment and HPV vaccination (i.e., the knowledge index score), questions were included regarding the etiology of cervical cancer. The present investigation operationalized the willingness to pay for the HPV vaccine as the inclination of unvaccinated females to obtain the HPV vaccine after being informed of its cost. The measurement of vaccine rejection was based on the response elicited by the following inquiry: “If the vaccine is not free, and you have to pay ‘out of pocket’ by yourself, will you vaccinate yourself and/or your daughter against HPV?” A follow-up question was used to assess the willingness to pay (WTP) of “vaccine acceptors.” The question read, “From the scale below, mark ‘*x*’ on the maximum amount you will pay (in Naira) to have yourself and/or your daughter vaccinated against HPV.” Individuals who responded negatively or reported a payment card value of zero were categorized as “vaccine rejecters,” whereas those who responded affirmatively and said a positive payment card value were classified as “vaccine acceptors.” The payment card provided a range of WTP values from zero to NGN 27,623.40, equivalent to USD 60. The highest price tendered corresponded to the overall market value of the vaccine in Nigeria. The perceived monetary benefit of the vaccine was deemed as the upper limit of their willingness to pay. The participants’ answers to the willingness to pay inquiries were classified into two groups: those who accepted the vaccine and those who declined it. The variable that became the dependent variable in the multivariate logistic regression was the response to the willingness to pay question.

Data collection procedure: The data collection was conducted between June 2022 and November 2022. It involved face-to-face interviews with eligible participants. All research assistants were trained before the commencement of the study on the research tools, interviewing skills, data management, and clarifications of ethical issues in research. The research assistants administered the questionnaires in Pidgin English or the local language for participants who could neither read nor write. The questionnaires were administered privately, and clarification and assistance were provided where necessary. The interviews took approximately 20 min to complete.

Statistical analysis: The data collected in the survey were analyzed using the SPSS version 27.0 (SPSS Inc. Chicago, IL, USA) statistical package. Descriptive statistics, including frequencies and percentages, were used to summarize the socio-demographic and health characteristics of the study’s participants. The means and standard deviations were used to summarize continuous variables such as age and income. The primary outcome variable was the willingness to pay for the HPV vaccine, which was measured as a binary variable (yes/no). To test the assumptions of a logistic regression, we checked for multicollinearity, linearity, and the normality of the residuals.

We assessed the presence of multicollinearity using the variance inflation factor (VIF), with a VIF value of 10 indicating high multicollinearity. We found that all independent variables had VIF values below the threshold, indicating no significant multicollinearity. We also examined the linearity assumption by plotting the residuals against the predicted values and examined the normality assumption by generating a standard probability plot of residuals. The residual plots showed no evidence of nonlinearity or non-normality, confirming that the assumptions of the logistic regression were met. We also conducted sensitivity and subgroup analyses to assess the robustness of the results using different beliefs and models. Finally, we used a logistic regression analysis to examine the factors associated with the willingness to pay for the HPV vaccine. The independent variables included in the regression model were socio-demographic variables, HIV-related health status variables, and knowledge about HPV and the HPV vaccine. The odds ratios (ORs) and 95% confidence intervals (CIs) were used to estimate the strengths and directions of the associations between the independent and outcome variables. A *p*-value of less than 0.05 was considered statistically significant. We also conducted sensitivity analyses to assess the robustness of the findings using different assumptions and models. Subgroup analyses examined the associations between the independent variables and the willingness to pay for the HPV vaccine by age group, income level, and educational level.

## 3. Ethical Considerations

This study was conducted following the ethical principles of the Declaration of Helsinki. Ethics approval was obtained from the Institutional Review Board of the Nigerian Institute of Medical Research, Lagos State (IRB-21-047). Before administering the questionnaire, the participants were given information sheets outlining the study’s objective and scope, which were duly explained to the participants in English or the local dialect (Yoruba/Pidgin). The participants were informed that participation in the study was voluntary and that they were free to withdraw from the study at any point without any consequences. The confidentiality and anonymity of the participants were ensured, and all data were kept confidential and were used only for research purposes. The participants were assured that participation or non-participation would not affect their access to healthcare services. In addition, participants who required psychological support after the study were referred to the appropriate healthcare professionals.

## 4. Results

Table 1 shows the socio-demographic characteristics of the 1371 participants in this study. The mean age of the participants was 43.2 ± 9.2. Most participants were married (809, 59%), and 668 (48.7%) had a tertiary level of education. Furthermore, 1008 (73.5%) of the participants were working. In addition, 587 (42.8%) of the participants had income levels that fell between NGN 18,000.00 (USD 40.0) and NGN 35,000.00 (USD 76.0).

Table 2 shows the knowledge of the HPV vaccine among the study’s participants. In total, 1085 (79.1%) said they had not heard of the HPV vaccine; 1092 (78.6%) participants said that if the vaccine exists, it will cure cervical cancer; 447 (32.6%) of the participants indicated that if the HPV vaccine has been taken, regular screening is still needed; 423 (30.9%) of the participants agreed that the HPV vaccine effectively prevents HPV infection, and 398 (29.0%) participants agreed that the HPV vaccine effectively prevents cervical cancer.

Table 3 presents the attitudes of women living with HIV in Lagos toward a willingness to pay for the HPV vaccine. In total, 937 (68.3%) said they were unwilling to pay for the HPV vaccine. When asked if they were willing to get their daughter vaccinated if the HPV vaccine was available, 762 (55.6%) said they were ready to get their daughter vaccinated. However, when asked if they were willing to pay for their daughter’s vaccine, only 455 (33.2%) agreed to pay for their daughter’s vaccine. When asked if they would allow all females around them to be vaccinated if vaccination was free, 1085 (79.1%) confirmed that if vaccination was free, all females around them would be vaccinated. Only 35 (2.6%) participants had been vaccinated with the HPV vaccine. The average amount the participants were willing to pay for the vaccine if it was available was NGN 3221.15 (USD 7.00) ± NGN 3963.95 (USD 8.61).

Table 4 shows the associations between individual factors and the willingness to pay for the HPV vaccine. The analysis revealed that the education level was significantly associated with the willingness to pay but only among participants who attended tertiary education (OR = 4.564, 95% CI: 1.860–11.164). The study also indicated that participants earning between NGN 51,000.00 and NGN 70,000.00 (USD 111 and USD 152) and those earning above NGN 100,000.00 (>USD 217) were willing to pay for the vaccine (OR = 2.178, 95% CI: 1.315–3.610 and OR = 3.673, 95% CI: 2.209–6.108, respectively).

However, after controlling for potential confounders and inter-relationships between factors, it was shown that participants who had attended tertiary education (aOR = 4.004; 95% CI: 1.623–9.877) and those that were earning more than NGN 100,000.00 (USD 217) (aOR = 2.468; 95% CI: 1.458–4.180) were willing to pay for HPV vaccination. Furthermore, knowledge of HPV (aOR = 2.270, 95% CI: 1.400–3.681) and cervical cancer (aOR = 4.241, 95% CI: 3.035–5.925) were statistically significantly associated with the participants’ willingness to get vaccinated. The knowledge of the HPV vaccine was initially significant (OR = 1.983; 95% CI: 1.454–2.704) but was not statistically significant after controlling for other factors (aOR = 1.284; 95% CI: 0.914–1.804).

Table 5 presents the associations between perceived screening benefits and the willingness to pay for the HPV vaccine. The analysis revealed a statistically significant relationship for each perceived benefit. However, after adjusting for other confounding factors, only participants with accurate knowledge of the HPV vaccine’s effectiveness against the development of cervical cancer (aOR = 1.856, 95% CI: 1.231–2.798), the effectiveness of the early detection of cervical cancer (aOR = 1.366, 95% CI: 1.049–1.779), and the significance of regular screening (aOR = 2.227, 95% CI: 1.609–3.082) as well as the knowledge that screening for cervical cancer in women with HIV can prevent the development of cancer (aOR = 2.009, 95% CI: 1.509–2.675) had statistically significant associations.

## 5. Discussion

This study investigated the willingness to pay for HPV vaccination among WLWH in Nigeria. Cervical cancer remains a significant public health concern in Nigeria. However, it is clear that HPV awareness is low, and specific knowledge was generally poor among WLWH in Nigeria.

The results of this study indicate a significant lack of knowledge related to HPV, cervical cancer, and the HPV vaccine among the participants, indicating a need for greater public awareness and education about these critical health issues. This outcome is similar to that of a study conducted among WLWH in Lagos, Nigeria, where 67.7% of the participants had never heard of HPV infection and only 22.3% knew about the HPV vaccine [28]. Moreover, a study conducted in the United States found that HPV vaccine knowledge and awareness were low among WLWH despite their dramatically increased risk of developing precancerous cervical lesions, cervical cancer, and other HPV-associated cancers [26,29].

The source of information about HPV and cervical cancer was primarily health workers, indicating the need for healthcare providers to play a more significant role in disseminating information about these health issues to the public. These findings are consistent with previous studies that have reported the critical role of health workers in improving the knowledge and awareness of HPV and cervical cancer [25,30].

This study revealed that most participants were unwilling to pay for the HPV vaccine, indicating that the cost is a substantial barrier, particularly for those with lower incomes. Only 3% of the participants had already received the HPV vaccine. The scenario above resembles a research endeavor in Vietnam, where individuals in the poor/near-poor household category have not received HPV vaccinations [31]. Multiple investigations across developing regions worldwide have identified elevated expenses as barriers to the willingness to accept and pay for vaccines [32,33]. The data suggest that women cannot access vital healthcare services due to financial limitations, even in areas where such services are available. Efforts aimed at promoting the HPV vaccine will need to carefully consider specific demographic patterns, including but not limited to educational attainment and income levels. Studies conducted on the willingness to pay for the HPV vaccine among Nigerian female undergraduates and the general population found that a higher percentage of participants were willing to pay for the HPV vaccine [15,34,35]. However, this contradicts this study’s findings, where most participants were unwilling to pay for the vaccine. This study’s finding might be because WLWH have other healthcare needs that take precedence over getting the HPV vaccine, such as managing their HIV medications or addressing other HIV-related complications.

This study also assessed the willingness of WLWH to pay for HPV vaccination for their daughters and females in their communities. The results showed that 55.6% of the participants were willing to vaccinate their daughters, while 66.8% were not willing to pay for the vaccine, indicating that the cost was a significant factor in their decision making. However, if the vaccine was free, 79.1% of the participants expressed their willingness to vaccinate their daughters and other females in their communities, suggesting potential demand for the vaccine. Another study found that 72.0% of parents were willing to vaccinate their daughters if the vaccine was free [36]. Studies conducted in other parts of the world have also shown that low-income individuals may be less willing to pay for the HPV vaccine due to financial constraints [10,33,37,38]. These findings suggest that the cost is a significant barrier to vaccine uptake, particularly in low-income settings. However, parents who perceive the vaccine as effective and necessary for their daughters are more willing to pay. Therefore, addressing the cost of the vaccine and educating parents on the benefits of HPV vaccination are crucial to improve vaccine uptake.

This study found that participants were willing to pay a meaningful amount of USD 7.46 ± USD 9.17 (NGN 3221.15 ± NGN 3963.95), which was significantly lower than the market cost. Therefore, reducing the vaccine cost could increase the willingness to pay and vaccine uptake, and a co-payment for HPV vaccination could be a viable option to augment the cost of vaccination in a government-funded vaccination scenario. This finding is consistent with previous studies conducted in Kenya and India, which identified costs as a significant barriers to HPV vaccination uptake and suggested strategies to reduce costs, such as government-sponsored programs and subsidies [39,40].

This study aimed to identify factors associated with the willingness to pay for the HPV vaccine. While the education level and income were not strongly associated with the willingness to pay for the HPV vaccine, this study found that participants who had attended tertiary education were more willing to pay for the vaccine than those who had not. This study suggests that higher education levels may contribute to determining an individual’s willingness to pay for the HPV vaccine. This study’s finding is consistent with previous research suggesting that education can influence health-related decision making [40,41]. Another study showed that knowledge of cervical cancer was significantly associated with the willingness to pay and that participants with higher education were willing to pay for the vaccine at a lower price than those with less secondary education [39,42]. Other studies conducted in the United States, Canada, and Ethiopia also highlighted the significance of various factors affecting vaccine uptake and the willingness to pay [43,44,45,46]. These factors included knowledge about HPV and its associated diseases, the perceived risk of HPV, the perceived effectiveness of the vaccine, and awareness of cervical cancer. Overall, the findings emphasized the importance of addressing these factors to increase vaccination rates and reduce the burden of HPV-related diseases.

A study conducted in Nigeria found that the cost and a lack of knowledge about cervical cancer and its prevention were significant barriers to vaccine uptake and willingness to pay [27]. This study’s findings support that study’s results, which showed that knowledge of HPV and the vaccine were significant predictors of willingness to pay [27]. Another study conducted in Nigeria also identified fear of side effects and a lack of trust in the safety and efficacy of the vaccine as essential factors contributing to low vaccine uptake and willingness to pay [28,47,48]. According to this study, the perceived benefits of screening are significantly related to the willingness to pay for the HPV vaccine. The results indicate that individuals with accurate knowledge about (1) the vaccine’s effectiveness in preventing cervical cancer, (2) the early detection of cervical cancer, (3) the importance of regular screening, and (4) the benefits of screening for cervical cancer in women with HIV are more willing to pay for the vaccine. These findings align with prior research suggesting that preventive health measures’ perceived benefits can impact healthcare decision making [49].

Furthermore, this study highlights that the knowledge of the HPV vaccine’s effectiveness in preventing cervical cancer is the most significant factor related to the willingness to pay for the vaccine. This finding suggests that increasing public awareness about the vaccine’s effectiveness in preventing cervical cancer could increase vaccine uptake.

Overall, the findings of this study and similar studies conducted in other countries emphasize the importance of addressing knowledge gaps, increasing awareness about the vaccine’s benefits, and managing cost barriers to improve vaccine uptake and the willingness to pay. In addition, targeted educational interventions and strategies to improve access to the vaccine and the affordability of the vaccine could help increase vaccine uptake and ultimately reduce the burden of cervical cancer.

This study has several limitations. First, the study was conducted at a single HIV treatment clinic in Nigeria and may not be generalizable to other populations or settings. Second, the study relied on self-reported data, which may be subject to social desirability or recall bias. Third, the study did not assess the knowledge and willingness to pay for the vaccine among women who were not receiving HIV treatment, and this may have biased the results. Fourth, the study did not explore other potential barriers to vaccine uptake, such as cultural or religious beliefs. Lastly, this study did not collect data on the history of violence or sexual abuse among the study participants, which could be a potential confounding factor for the willingness to pay for the HPV vaccine. Therefore, we cannot rule out the possibility that past experiences of violence or sexual abuse may have influenced the results of our analysis.

In terms of prospects, this study highlights the need for effective strategies to increase awareness and access to the HPV vaccine among women living with HIV in Nigeria. Community outreach programs, school-based education programs, and targeted campaigns for women and parents could be effective strategies to increase vaccine uptake. Additionally, further research is necessary to explore other factors that may influence the willingness to pay for the HPV vaccine in Nigeria and to identify potential barriers to vaccine uptake that were not explored in this study. Future studies should consider collecting such data to better understand the relationship between past experiences of violence or sexual abuse and the willingness to pay for the HPV vaccine among women living with HIV.

## 6. Conclusions

This study highlights the low levels of knowledge about HPV, cervical cancer, and the HPV vaccine among WLWH in Nigeria, resulting in 66.8% of participants being unwilling to pay for the vaccine. The study emphasizes the importance of increasing education and awareness about the vaccine’s significance in preventing cervical cancer, with health workers playing a crucial role in providing information. Factors associated with the willingness to pay, such as income, knowledge of HPV and the vaccine, awareness of cervical cancer, and the health belief model of a perceived screening benefit, were also identified. The study’s findings could aid in developing effective strategies to increase vaccine uptake in Nigeria, such as community outreach and school-based education programs. Further research is necessary to explore additional factors influencing the willingness to pay for the vaccine. This study recommends that the Nigerian government and healthcare providers prioritize increasing awareness and education about the HPV vaccine and cervical cancer prevention among WLWH.

## Figures and Tables

**Table 1 vaccines-11-00928-t001:** Participants’ socio-demographic characteristics (N = 1371).

Socio-Demographic Characteristics	N (%)
**Age (Mean ± SD)**	43.2 ± 9.2
**Age group**	
≤30	117 (8.5)
31–40	409 (29.8)
41–50	578 (42.2)
51–60	225 (16.5)
>60	41 (3.0)
**Ethnicity**	
Igbo	545 (39.8)
Yoruba	499 (36.4)
Hausa	29 (2.1)
Others	298 (21.7)
**Education level**	
No Education	24 (1.8)
Primary	120 (8.8)
Secondary	559 (40.8)
Tertiary	668 (48.7)
**Profession**	
Unemployed	56 (4.1)
Self-employed	1016 (74.1)
Professional	240 (17,5)
Civil servant	59 (4.3)
**Marital status**	
Single	256 (18.7)
Married	809 (59)
Separated	85 (6.2)
Divorced	20 (1.5)
Widowed	201 (14.7)
**Income (NGN)**	
<18,000	358 (26.1)
18,000–35,000	587 (42.8)
36,000–50,000	165 (12)
51,000–70,000	82 (6)
71,000–100,000	77 (5.6)
>100,000	102 (7.4)
**Number of sexual partners**	
None	1099 (80.2)
1	242 (17.7)
<3	28 (2.0)
>3	2 (0.1)
**Source of Income**	
Family	329 (24)
Wages	667 (48.7)
Salary	375 (27.4)

**Table 2 vaccines-11-00928-t002:** Knowledge of the HPV vaccine.

Knowledge/Belief	No N (%)	Yes N (%)
**Have you heard of the HPV vaccine?**	1085 (79.1)	86 (20.9)
**Does the HPV vaccine cure cancer?**	279 (20.4)	1092 (79.6)
**Is regular screening for cancer still needed even though you have been vaccinated against HPV?**	924 (67.4)	447 (32.6)
**Is the HPV vaccine highly effective in preventing HPV infection?**	948 (69.1)	423 (30.9)
**Is the HPV vaccine highly effective in preventing cervical cancer?**	973 (71.0)	398 (29.0)

**Table 3 vaccines-11-00928-t003:** Attitude/willingness to pay for the HPV vaccine.

Attitude/Willingness	No N (%)	Yes N (%)
**Are you willing to pay for the HPV vaccine?**	937 (68.3)	434 (31.7)
**Are you willing you get your daughter vaccinated?**	609 (44.4)	762 (55.6)
**Would you be willing to pay for your daughter’s vaccine?**	916 (66.8)	455 (33.2)
**If vaccination was free, would you allow all females around you to be vaccinated?**	286 (20.9)	1085 (79.1)
**Have you been vaccinated against HPV?**	1336 (97.4)	35 (2.6)
**Amount willing to pay (Mean ± SD)**	3221.15 ± 3963.950	

**Table 4 vaccines-11-00928-t004:** Factors associated with the willingness to pay for the vaccine.

Variables	Category	Crude OR 95% CI (Lower Bound–Upper Bound)	*p*-Value	Adjusted OR 95% CI (Lower Bound–Upper Bound)	*p* Value
**Education Level**	No education	Ref		Ref	
	Primary	1.308 (0.502–3.404)	0.583	1.273 (0.488–3.321)	0.621
	Secondary	2.386 (0.974–5.842)	0.057	2.266 (0.923–5.564)	0.074
	Tertiary	4.564 (1.860–11.164)	0.001 **	4.004 (1.623–9.877)	0.003 **
**Income (NGN)**	<18,000	Ref		Ref	
	18,000–35,000	1.282 (0.985–1.668)	0.065	1.183 (0.904–1.549)	0.221
	36,000–50,000	1.283 (0.886–1.858)	0.186	0.972 (0.661–1.429)	0.884
	51,000–70,000	2.178 (1.315–3.610)	0.003 **	1.649 (0.983–2.766)	0.058
	71,000–100,000	1.504 (0.914–0.475)	0.109	1.047 (0.625–1.755)	0.861
	>100,000	3.673 (2.209–6.108)	0.000 **	2.468 (1.458–4.180)	0.001 **
**Knowledge of HPV**	Poor knowledge	Ref		Ref	
	Good knowledge	3.899 (2.489–6.108)	0.000 **	2.270 (1.400–3.681)	0.001 **
**Knowledge of the HPV Vaccine**	Poor knowledge	Ref		Ref	
	Good knowledge	1.983 (1.454–2.704)	0.000 **	1.284 (0.914–1.804)	0.149
**Knowledge of cervical cancer**	Poor knowledge	Ref		Ref	
	Good knowledge	5.061 (3.657–7.004)	0.000 **	4.241 (3.035–5.925)	<0.001 **

** *Statistically significant*.

**Table 5 vaccines-11-00928-t005:** Health beliefs associated with the willingness to pay.

Variables	Category	Crude OR 95% CI (Lower Bound–Upper Bound)	*p*-Value	Adjusted OR 95% CI (Lower Bound–Upper Bound)	*p* Value
** *Perceived screening benefit* **					
**Screening for cervical cancer in women with HIV can prevent the development of cancer.**	No	Ref		Ref	
	Yes	3.907 (3.056–4.994)	0.000 **	2.009 (1.509–2.675)	<0.001 **
**Is regular screening for cervical cancer needed even though you have been vaccinated against HPV?**	No	Ref		Ref	
	Yes	4.991 (3.832–6.502)	0.000 **	2.227 (1.609–3.082)	<0.001 **
**Early detection of cervical cancer can increase survival.**	No	Ref		Ref	
	Yes	2.422 (1.946–3.016)	0.000 **	1.366 (1.049–1.779)	0.021 **
** *Susceptibility to cervical cancer and HPV infection* **					
**The germ that can cause cervical cancer can be transmitted through sexual intercourse.**	No	Ref		Ref	
	Yes	1.784 (1.430–2.225)	0.000 **	0.916 (0.695–1.206)	0.530
**HPV is transmitted during sexual intercourse.**	No	Ref		Ref	
	Yes	3.093 (2.407–3.975)	0.000 **	1.287 (0.944–1.755)	0.111
** *Perceived benefit of the vaccine* **					
**Are HPV vaccines highly effective for HPV infection?**	No	Ref		Ref	
	Yes	4.316 (3.314–5.620)	0.000 **	1.221 (0.808–1.846)	0.343
**Are HPV vaccines highly effective against cervical cancer?**	No	Ref		Ref	
	Yes	4.644 (3.528–6.111)	0.000 **	1.856 (1.231–2.798)	0.003 **

** *Statistically significant*.

## Data Availability

The data used to support the findings of this study are available from the corresponding author upon request.

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
