# Peer review of "Willingness to Pay for HPV Vaccine among Women Living with HIV in Nigeria"

_vaccines, 2023, doi:10.3390/vaccines11050928_

Round 1

Reviewer 1 Report

This is the cross-sectional study conducted on about 1371 participants for six months from June 2022 to Nov2022 in the Nigerian population. The current study represents an original piece of work and is well-written. The authors of this study elucidate the impediments that impede the HPV vaccination uptake in women with HIV.

Minor Comments:

Line Number67: Mention the abbreviation of WLHIV

Line128: WTP stands for???

In table 1: Authors should mention the type of working status???

Author Response

Comments and Suggestions for Authors

This is the cross-sectional study conducted on about 1371 participants for six months from June 2022 to Nov2022 in the Nigerian population. The current study represents an original piece of work and is well-written. The authors of this study elucidate the impediments that impede the HPV vaccination uptake in women with HIV.

 Minor Comments:

Line Number67: Mention the abbreviation of WLHIV

Line128: WTP stands for???

In table 1: Authors should mention the type of working status???

Thank you for your feedback, and constructive review. Please, find below my responses.

Minor Comment 1: Line Number67 - Mention the abbreviation of WLHIV

Response 1: The abbreviation WLHIV has been corrected to WLWH. The first definition of WLWH has been defined in Lines 44 – 45. WLWH stands for “Women living with HIV.”

Minor Comment 2: Line 128 – WTP stands for???

Response 2: The abbreviation of WTP has been defined in Line 135 as “Willingness to pay.”

Minor Comment 3: Authors should mention the type of working status???

Response 3: Thank you for this observation. We have deleted the working status. We realized it will be better to focus on the profession which has been categorized into unemployed; self-employed; professional and civil servant.

Reviewer 2 Report

Dear Editor,
I really appreciate the opportunity to review the manuscript vaccines-2326469 entitled:
"Willingness to pay for HPV vaccine among women living with HIV in Nigeria"

I commend the authors for describing this critical and timely issue. The paper is interesting and well-written; however, I would like to highlight some issues that merit revision:

The article is particularly interesting and well-written; it does not appear clear from the manuscript what influence past violence or sexual abuse experienced has on the scope analyzed. I beg the authors to supplement, or if the data is not available to add it as a limitation

Author Response

Dear Editor,

I really appreciate the opportunity to review the manuscript vaccines-2326469 entitled:

"Willingness to pay for HPV vaccine among women living with HIV in Nigeria"

I commend the authors for describing this critical and timely issue. The paper is interesting and well-written; however, I would like to highlight some issues that merit revision:

The article is particularly interesting and well-written; it does not appear clear from the manuscript what influence past violence or sexual abuse experienced has on the scope analyzed. I beg the authors to supplement, or if the data is not available to add it as a limitation

Thank you for your feedback, and constructive review. Please, find below my responses.

Past violence or sexual abuse experienced by women living with HIV (WLWH) may impact their willingness to pay for the HPV vaccine in several ways which could lead to a lower likelihood of WLWH being aware of the HPV vaccine or being willing to pay for it. Unfortunately, this study did not collect data on the history of violence or sexual abuse among the study participants, which could be a potential confounding factor for willingness to pay for the HPV vaccine.

It has been included as part of the study limitation and future prospects.

These changes can be viewed from Line 334-350

“The study has several limitations. First, the study was conducted at a single HIV treatment clinic in Nigeria and may not be generalizable to other populations or settings. Second, the study relied on self-reported data, which may be subject to social desirability bias or recall bias. Third, the study did not assess the knowledge and willingness to pay for the vaccine among women who were not receiving HIV treatment, and this may have biased the results. Fourth, the study did not explore other potential barriers to vaccine uptake, such as cultural or religious beliefs. Lastly, this study did not collect data on the history of violence or sexual abuse among the study participants, which could be a potential confounding factor for willingness to pay for the HPV vaccine. Therefore, we cannot rule out the possibility that past experiences of violence or sexual abuse may have influenced the results of our analysis.

In terms of future prospects, the study highlights the need for effective strategies to increase awareness and access to the HPV vaccine among women living with HIV in Nigeria. Community outreach programs, school-based education programs, and targeted campaigns for women and parents could be effective strategies to increase vaccine uptake. Additionally, further research is necessary to explore other factors that may influence willingness to pay for the HPV vaccine in Nigeria and to identify potential barriers to vaccine uptake that were not explored in this study. Future studies should consider collecting such data to better understand the relationship between past experiences of violence or sexual abuse and willingness to pay for the HPV vaccine among women living with HIV”.

Reviewer 3 Report

The problem question is what are peculiarities of willingness to pay for HPV vaccine among women living with HIV in Nigeria?

The topic of the manuscript is relatively original and is within the scope of the Journal and could be valuable to the scientific audience. While the HPV vaccine is a promising solution to reduce the incidence of cervical cancer, the question of whether the HPV vaccine would be used among HIV-positive women in Nigeria was not known, and was sought to be answered.

This study bridging the gap because there were examined different factors associated with the willingness to pay for the HPV vaccine.

The title of the article is accurate. Abstract reflects the work done and the conclusions drawn.

Some clarifications are however needed:

All abbreviations must be explained (WLWH, WTP).

The aim of the study must be clearly stated in the last paragraph of the Introduction section.

Research hypothesis is missing. Authors should provide justification for the research hypothesis.

Please address the following issues:

Have the assumptions for logistic regression been tested and met? For instance, absence of multicollinearity, and lack of strongly influential outliers?

 I do not understand what "₦3221.15" (line 203) means.

 I think that P-values cannot be reported as "P = 0.000". This is statistically incorrect. P-values must be reported as "P < 0.001".

 I suppose that the limitations of the study must be defined and future prospects should be described.

 Conclusions can be improved. Conclusion “The majority of participants were not willing to pay for the HPV vaccine, likely due to inadequate 327 awareness and knowledge about it” is not correct. What does it mean majority? Please indicate percents.

The references are not correctly presented. 

TO SUM UP I think the author(s) need to make the recommended corrections.

Author Response

The problem question is what are peculiarities of willingness to pay for HPV vaccine among women living with HIV in Nigeria?

The topic of the manuscript is relatively original and is within the scope of the Journal and could be valuable to the scientific audience. While the HPV vaccine is a promising solution to reduce the incidence of cervical cancer, the question of whether the HPV vaccine would be used among HIV-positive women in Nigeria was not known, and was sought to be answered.

This study bridging the gap because there were examined different factors associated with the willingness to pay for the HPV vaccine.

The title of the article is accurate. Abstract reflects the work done and the conclusions drawn.

Some clarifications are however needed:

  • All abbreviations must be explained (WLWH, WTP).
  • The aim of the study must be clearly stated in the last paragraph of the Introduction section.
  • Research hypothesis is missing. Authors should provide justification for the research hypothesis.
  • Please address the following issues:
  • Have the assumptions for logistic regression been tested and met? For instance, absence of multicollinearity, and lack of strongly influential outliers?
  • I do not understand what "₦3221.15" (line 203) means.
  • I think that P-values cannot be reported as "P = 0.000". This is statistically incorrect. P-values must be reported as "P < 0.001".
  • I suppose that the limitations of the study must be defined and future prospects should be described.
  • Conclusions can be improved. Conclusion “The majority of participants were not willing to pay for the HPV vaccine, likely due to inadequate 327 awareness and knowledge about it” is not correct. What does it mean majority? Please indicate percents.
  • The references are not correctly presented.

TO SUM UP I think the author(s) need to make the recommended corrections.

Thank you for your feedback, and constructive review. Please, find below my responses.

  1. All abbreviations must be explained (WLWH, WTP)

The full definition of the abbreviations WLWH and WTP have been explained in Lines 41 and 132 respectively.

WLWH – Women living with HIV

WTP – Willingness to Pay

  1. The aim of the study must be clearly stated in the last paragraph of the Introduction section.

The aim of the study has been properly stated in Lines 93 and 95.

“the study aimed to identify factors associated with willingness to pay for the HPV vaccine among women living with HIV in Nigeria. The study also sought to assess the knowledge of HPV, cervical cancer, and the HPV vaccine among this population”.

  1. Have the assumptions for logistic regression been tested and met? For instance, absence of multicollinearity, and lack of strongly influential outliers?

Thank you for this crucial observation. The missing information has been included in Lines 154-161.

“To test the assumptions for logistic regression, we checked for multicollinearity, linearity, and normality of residuals. We assessed the presence of multicollinearity using the variance inflation factor (VIF), with a VIF value of 10 indicating high multicollinearity. We found that all independent variables had VIF values below the threshold, indicating no significant multicollinearity. We also examined the linearity assumption by plotting the residuals against the predicted values, and the normality assumption by generating a normal probability plot of residuals. The residual plots showed no evidence of nonlinearity or non-normality, confirming that the assumptions for logistic regression were met. We also conducted sensitivity and subgroup analyses to assess the robustness of the results to different assumptions and models.”

  1. I do not understand what "₦3221.15" (line 203) means.

Thank you for the observation. It is now written as “3221.15 NGN (7.00 USD) ± 3963.95 NGN (8.61 USD)”. Line 212 indicates how much the participants were willing to pay in Nigerian Naira (NGN) and the equivalence in USD.

NGN – Nigerian Naira

USD – United States Dollar

  1. I think that P-values cannot be reported as "P = 0.000". This is statistically incorrect. P-values must be reported as "P < 0.001".

Yes, this is correct. P-values should be reported as "P < 0.001" instead of "P = 0.000" because a p-value cannot be exactly zero, but can be very small.

I have changed the results indicating P=0.000 to <0.001**

  1. I suppose that the limitations of the study must be defined and future prospects should be described.

I have included the study limitations and future prospects in Lines 333 -349.

“The study has several limitations. First, the study was conducted at a single HIV treatment clinic in Nigeria and may not be generalizable to other populations or settings. Second, the study relied on self-reported data, which may be subject to social desirability bias or recall bias. Third, the study did not assess the knowledge and willingness to pay for the vaccine among women who were not receiving HIV treatment, and this may have biased the results. Fourth, the study did not explore other potential barriers to vaccine uptake, such as cultural or religious beliefs. Lastly, this study did not collect data on the history of violence or sexual abuse among the study participants, which could be a potential confounding factor for willingness to pay for the HPV vaccine. Therefore, we cannot rule out the possibility that past experiences of violence or sexual abuse may have influenced the results of our analysis.

In terms of future prospects, the study highlights the need for effective strategies to increase awareness and access to the HPV vaccine among women living with HIV in Nigeria. Community outreach programs, school-based education programs, and targeted campaigns for women and parents could be effective strategies to increase vaccine uptake. Additionally, further research is necessary to explore other factors that may influence willingness to pay for the HPV vaccine in Nigeria and to identify potential barriers to vaccine uptake that were not explored in this study. Future studies should consider collecting such data to better understand the relationship between past experiences of violence or sexual abuse and willingness to pay for the HPV vaccine among women living with HIV”.

  1. Conclusions can be improved. Conclusion “The majority of participants were not willing to pay for the HPV vaccine, likely due to inadequate 327 awareness and knowledge about it” is not correct. What does it mean majority? Please indicate percents.

The study conclusion has been reworded for clarity in Lines 351 -362.

“This study highlights the low levels of knowledge about HPV, cervical cancer, and the HPV vaccine among WLWH in Nigeria, resulting in 66.8% of participants being unwilling to pay for the vaccine. The study emphasizes the importance of increasing education and awareness about the vaccine's significance in preventing cervical cancer, with health workers playing a crucial role in providing information. Factors associated with willingness to pay, such as income, knowledge of HPV and the vaccine, awareness of cervical cancer, and the health belief model of perceived screening benefit, were also identified. The study's findings could aid in the development of effective strategies to increase vaccine uptake in Nigeria, such as community outreach and school-based education programs. Further research is necessary to explore additional factors that influence willingness to pay for the vaccine, and it is recommended that the Nigerian government and healthcare providers prioritize increasing awareness and education about the HPV vaccine and cervical cancer prevention among WLWH”.

  1. The references are not correctly presented.

Thank you. We have adjusted the references with poor presentation.

Round 2

Reviewer 3 Report

The authors have satisfactorily addressed each of my concerns.

My central concern related to testing of the assumptions for logistic regression. This has been addressed and is fine now.

Limitations future prospects: this has been effectively addressed. The authors added a section to highlight more clearly limitations and suggestions for future research.